# Dilated Skip Convolution for Facial Landmark Detection

**DOI:** 10.3390/s19245350

**Published:** 2019-12-04

**Authors:** Seyha Chim, Jin-Gu Lee, Ho-Hyun Park

**Affiliations:** School of Electrical and Electronics Engineering, Chung-Ang University, 84 Heukseok-ro, Dongjak-gu, Seoul 06974, Korea; seyhachim@cau.ac.kr (S.C.); dlwlsrn21@cau.ac.kr (J.-G.L.)

**Keywords:** face landmark detection, fully convolutional DenseNets, skip-connections, dilated convolutions

## Abstract

Facial landmark detection has gained enormous interest for face-related applications due to its success in facial analysis tasks such as facial recognition, cartoon generation, face tracking and facial expression analysis. Many studies have been proposed and implemented to deal with the challenging problems of localizing facial landmarks from given images, including large appearance variations and partial occlusion. Studies have differed in the way they use the facial appearances and shape information of input images. In our work, we consider facial information within both global and local contexts. We aim to obtain local pixel-level accuracy for local-context information in the first stage and integrate this with knowledge of spatial relationships between each key point in a whole image for global-context information in the second stage. Thus, the pipeline of our architecture consists of two main components: (1) a deep network for local-context subnet that generates detection heatmaps via fully convolutional DenseNets with additional kernel convolution filters and (2) a dilated skip convolution subnet—a combination of dilated convolutions and skip-connections networks—that are in charge of robustly refining the local appearance heatmaps. Through this proposed architecture, we demonstrate that our approach achieves state-of-the-art performance on challenging datasets—including LFPW, HELEN, 300W and AFLW2000-3D—by leveraging fully convolutional DenseNets, skip-connections and dilated convolution architecture without further post-processing.

## 1. Introduction

In computer vision, facial landmark detection is known as face alignment and is a crucial part of face recognition operations. Its algorithms attempt to predict the locations of the fiducial facial landmark coordinates that vary owing to head movements and facial expressions. These landmarks are located at major parts of the face, such as the contours, tip of the nose, chin, eyes, corners of the mouth (see [1] in review). Facial landmark detection has sparked much interest recently as it is a prerequisite in many computer vision applications, including facial recognition [2], facial emotion recognition [3,4], face morphing [2,5], 3D face modelling [6] and human-computer interactions [7]. In recent years, considerable research works [8,9,10] have developed remarkable networks to predict facial landmark location more accurately even under challenging conditions, such as large appearance variations, facial occlusion and difficult illumination. Facial landmark detection is classified into three types of methods: holistic, constrained local model (CLM), and regression-based. Among these, regression-based approaches [5,11] have demonstrated superiority in both efficiency and accuracy, even in challenging scenarios. Regression-based methods contain two stages: early and updated. The inceptive key points are located on the predicted face shape in the early stage and gradually refined in the updated stage. However, [1] points out two main issues of this approach. The first issue is the sensitivity of the face detector. Commonly, the face is initially determined by the face bounding box. In the case it fails to detect the face in the first place, the accuracy also declines. Another issue is that the algorithms apply a fixed number of predictions, so it is impossible to judge the quality of the landmark prediction and adapt the necessary stages for different image tests.

Before the success of deep learning [9,12] for computer vision problems, [13] used a scale-invariant feature transform (SIFT) algorithm to learn appearance models from current landmarks. The algorithm iteratively regresses the models until the convergence criteria are reached. Recently, discriminative models such as convolutional neural networks (CNNs) and recurrent neural networks (RNNs) have dominated the field of facial landmark detection. Deep learning based models have been shown to outperform SIFT based models, which use hand-crafted features, for many vision tasks [14]. Hierarchical deep learning structures, in particular CNNs, can generate feature descriptors that capture more complex image characteristics and learn task specific features. In contrast, SIFT is not robust to non-linear transformations, particularly where SIFT cannot match sufficient feature points. It is unsuitable for data with large intra-class shape variations. Consequently, deep learning has attracted more attention than SIFT for computer vision applications. In early research [15], a probabilistic deep model for facial landmark detection that captured facial shape variations caused by poses and expressions was used. Also, [16] proposed to extract shape-indexed deep features from fully convolutional networks (FCNs) and refine the landmark locations recurrently via recurrent attentive-refinement (RAR) networks. In the early stage of [16]’s study, the network employed direct methods to regress key points directly on given images that are highly non-linear and difficult to estimate key point positions.

The research in [17] argues that learning indirectly to extract discriminative features from images yields more advantages over direct mapping. Accordingly, [17] applies an indirect prediction framework based on heatmap regression at individual body key points over the raw image. Furthermore, [17] mentions that adding several large convolutions (e.g., 13 × 13 kernel convolution) would improve estimation performance, although this increases the number of parameters and makes optimizations more difficult.

To address this problem, [18] pursued dilated convolutions that increase the effective receptive fields without introducing additional parameters. Intuitively, applying heatmap regression methods in a network of large convolutional kernels and deeper models enhances the performance of overall networks. Thus, we propose a deep end-to-end model which leverages fully convolutional DenseNets (FC-DenseNets) [19] that use heatmap regression to learn deep feature maps from the given image. Moreover, inspired by [18], we carefully designed a network that can extract more complex data dependencies by building extra skip-connections in the stacked dilated convolutions network. In doing so, we expect that our network will obtain different sizes of receptive fields and informative feature maps, which will boost prediction accuracy.

The main contributions of this work are as follows:To the best of our knowledge, this is the first work to exploit FC-DenseNets as a local detector with a heatmap regression to predict dense heatmaps from the given image.We designed a thorough dilated skip convolution (DSC) network that can refine the estimated heatmaps of the facial key points by combining a stack of dilated convolutions and a skip-connections method.We developed a robust method to estimate the initial facial shape to work in challenging conditions.We evaluated our framework’s performance with other state-of-the-art networks on LFPW [20], HELEN [21], 300 W [22] and AFLW2000-3D [23] datasets.

The rest of this paper is organized as follows. First, a summary of our paper’s relevant works is given in Section 2. Next, we present in detail our proposed methodology in Section 2. Then, the results of our experiments are presented in Section 4. Finally, the conclusions are drawn in Section 5.

## 2. Related Works

Facial landmark detection is divided into three types of methods: holistic, constrained local model (CLM) and regression-based. Holistic methods build a global model to learn the facial appearance and obtain shape information during training to estimate the best fits of any given test face image during testing via the model parameters. CLM methods use independent local appearance information around each landmark combined with a global face shape model for facial landmark detection, outperforming holistic methods for capturing illumination and occlusion. Unlike the first two methods, which build a global shape model, regression-based methods directly map the local facial appearance and regress the landmark locations between individual inputs and outputs.

### 2.1. Regression-Based Methods

Regression based methods have recently demonstrated outstanding performance compared with holistic and CLM methods. Regression based methods effectively build a parametric face shape or appearance model to extract feature maps from an image and infer a facial shape. Regression functions initially focus on holistic picture details, subsequently updating those features using finer image details to provide more accurate predictions. Using typical approaches, [5,11] proposed a regression function to predict landmark coordinates from shape indexed feature maps from the input image. Subsequently, [24] proposed a combined regression network to initially detect facial landmarks and then refine landmark locations using their scoremaps at progressively finer detail; and [25] proposed a cascade stacked auto-encoder network to produce finer images from low resolution input images; and [26] proposed multiple cascaded regressors to learn discriminative features around each facial landmark. Extending this early work, [27] proposed a two-step facial segmentation network to estimate head pose, gender and expression. The system first segmented face images into semantically small regions, for example hair, skin, nose, eyes, background, mouth and so forth.; and then classified thee regions using support vector machines (SVMs). The [27] process is effectively an extended version of the FASSEG dataset [28]. Rather than directly manipulating images in the spatial domain, [3,4] represented images as signals in the frequency domain with high time-frequency resolution. They then extracted useful feature maps from the decomposed image and employed supervised learning algorithms to classify facial expressions in the images. Ref. [3] applied stationary wavelet entropy to extract features in the frequency domain followed by a single hidden layer feedforward neural network, using the Jaya algorithm, a gradient-free optimizer. Similarly, [4] proposed biorthogonal wavelet entropy to extract multi-scale information and employed fuzzy multiclass SVM classifiers. Heatmap regression has also been used to estimate human pose, [29,30] and detect facial landmarks, [8,9,10]. Ref. [29] employed multiple regressors to predict human poses. The first regressor crops the input image to focus only on the human torso, reducing required computational resources for background analysis. [29] used subsequent regressors to roughly estimate joint locations and then crop joint centers and repeatedly regress the image. This not only considerably reduces the number of network parameters but also increases prediction accuracy since there is no information loss compared with using pooling layers to reduce data size. Ref. [30] proposed a stacked hourglass network to capture information from local to global scale and hence enable the network to learn spatial relationships between joints. Similarly, [8] cascaded four stacked hourglass networks in heatmaps regression to extract discriminative features from images, which were subsequently used to detect facial landmarks. Ref. [9] proposed a three step regression network based on convolutional response maps and component based models to robustly detect facial landmarks. Ref. [10] proposed combining heatmap and coordination contextual information into a feature representation that was subsequently refined by an arbitrary convolutional neural network (CNN) model.

### 2.2. Fully Convolutional Heatmap Regression Methods

Early methods used heatmap regression as an approach for 2D pose estimation [5,8,17,31]. Unlike the holistic regression methods, heatmap regression methods have the benefit of providing higher output resolutions that assist in accurately localizing the key points in the image via per-pixel predictions. To leverage this advantage, [17,31] regress a heatmap over the image for each key point and then obtain the key point position as a mode in this heatmap. Ref. [31] presents a convolutional network architecture incorporating motion features as a cue for body part localization and [17] proposes a CNN model to predict 2D human body poses in an image. The model regresses a heatmap representation for each body key point, learning and representing both partial appearances and the context of those partial configurations. In contrast, [5,8] exploit FCNs to estimate dense heatmaps for facial landmark detection. Ref. [5] proposes a two-step detection followed by a regression network to create the detection score map for each landmark, whereas [8] uses a stacked hourglass network for 2D and 3D face alignment.

#### Fully Convolutional DenseNets

Densely connected convolutional networks (DenseNets) [32] introduce a connectivity pattern that proves the gradient-vanishing problem can be solved even though the depth of CNN is increased. At the same time, the number of parameters can be reduced by connecting each layer with additional inputs from all preceding layers and reusing its feature maps in all subsequent layers. Recently, FC-DenseNets [19] extend DenseNets to be a fully convolutional network that achieves state-of-the-art results by tackling problem semantics with image segmentation. The resulting network is a deep network between 56 and 103 layers that has very few parameters. The goal of FC-DenseNets is to further exploit feature reuse by extending the more sophisticated DenseNets architecture while avoiding feature explosion at the upsampling path of the network. To recover the input spatial resolution, FC-DenseNets implicitly inherit the advantages of DenseNets that use pooling operations and dense blocks (DBs) to perform iterative concatenation of feature maps. The feature maps have a sufficiently large amount of detailed spatial information. To some extent, heatmap regression through FC-DenseNets is especially useful for multiple outputs per input (e.g., multiple faces).

FC-DenseNets are constructed from two symmetric parts where the downsampling part is an exact mirror of the upsampling part as shown in Figure 1. FC-DenseNets consist of 11 DBs: 5 DBs in the downsampling part followed by its own transitions down (TD), 5 DBs in the upsampling part followed by its own transitions up (TU) and one DB in the middle and so-called “bottleneck”. Each DB layer is composed of dense layers followed by batch normalization [33] and ReLU [34]. The solid line in Figure 1 represents the connection between each dense block of the fully convolutional DenseNet (FC-DenseNet), which passes output feature maps forward from one dense block to the next, whereas the dashed line indicates skip connections between FC-DenseNet downsampling and upsampling paths. The overall FC-DenseNet goal is to capture spatially detailed information from the downsampling path and recover it in the upsampling path by reusing the features maps. The last layer in the network is a 1×1 convolution followed by a softmax nonlinearity function to predict the class label.

### 2.3. Dilated Convolutions

Dilated (or atrous) convolutions have been widely utilized for various dense prediction and generation applications. As indicated in Reference [35], dilated convolutions enlarge exponentially receptive fields without loss of resolution or convergence while the number of parameters grows linearly. Larger kernel receptive fields can increase network capability to capture spatial context, which is beneficial to reconstruct large and complex edge structures. However, ordinary convolutions require a large number of parameters to expand their receptive fields. In contrast to ordinary convolutions, dilated convolution has zero-padding inside its kernels, injecting zeros into defined gaps to expand receptive field size, as shown in Figure 2. Thus, dilated convolutions can view larger input image portions without requiring a pooling layer, resulting in no spatial dimension loss and reduced computational time.

For semantic segmentation tasks, Reference [35] presents a new convolutional architecture that fully exploits dilated convolutions for multi-scale context aggregation. Reference [36] proposes two simple, yet effective, gridding methods by studying the decomposition of dilated convolutions. In these studies, dilated convolutions replace the need to upsample parts to keep the output resolutions the same as the input size. For other tasks such as audio generation [37], video modeling [38] and machine translation [39], dilated convolutions are used to capture global views of inputs with fewer parameters. WaveNet [37] was proposed by Google DeepMind and employs dilated convolutions to generate and recognize speech from raw audio waveforms. The dilation factor in Reference [37] is doubled, starting from 1 to a fixed factor number for every forward layer; then, the pattern is repeated.

Figure 2 illustrates how dilated convolutions enlarge the receptive fields by altering dilation factors (*d*). When dilation factors are increased exponentially, the gap pixels between the original kernel elements get progressively wider; this causes the receptive field to expand. In Figure 2a, a dilation factor of 1 (1-Dilated convolution) is performed in a dense 3×3 field on a feature map. We observed that the 1-Dilated convolution is the same as the 3×3 standard convolution filter. When the dilation factor is set to 2 as shown in Figure 2b, the region of the receptive field is increased dramatically to 7×7 pixels. The same occurs in Figure 2c when the dilation factor is changed to 4 and the receptive field is 15×15 pixels. In Figure 2, the group of red boxes is a 3×3 input filter that captures the receptive field (represented by the gray area) and the blue number indicates the meaning of the dilation factors that are applied to the kernels. The most important factor is the number of space pixels between the original kernel elements. In our work, we stack 7 dilated convolution layers with different dilation factors together to perceive a wider range for capturing global contexts of input feature maps.

## 3. Methods

The proposed facial landmark detection architecture is illustrated in Figure 3. We divide our approach into two connected sub-parts: the local appearance initialization (LAI) subnet and the dilated skip convolution (DSC) subnet for shape refinement. LAI pursues a heatmap regression approach convolved with kernel convolution to serve as a local detector of facial landmarks and the DSC subnet is designed to refine the local prediction of the first subnet.

### 3.1. Local Appearance Initialization Networks

It is well known that facial landmark detection uses single specific pixel location data p(x,y) as a training label where *x* and *y* are pixel coordinates in 2D images. However, using the training label data as a single-pixel point p(x,y) is inefficient for learning features from the input data. Even though the model returns a result close to the ground-truth pixel, a result that does not comply with the exact pixel location data p(x,y) may be considered wrong; as a result, the model may search for another pattern despite being close to the answer.

Recently, Gaussian distribution has come into play for manipulating the training label into a Gaussian heatmap label. It modifies the training label, not as a single specific point p(x,y) but rather as probabilities near the given training label pixel point. References [24,40] present several successful heatmap implementations in facial alignment. As presented by both papers, using heatmaps as a training label allows the network to learn faster. Furthermore, heatmaps demonstrate how the network is thinking during training since heatmaps are more visible to the naked eye. The correct point will have the highest probability in the distribution, whereas the neighboring pixels close to the correct pixel will also have high probabilities but not as high as that of the correct pixel. In Equation (Equation 1), the value of ${\hat{p}}_{i}$ will let the network know whether or not it is making a guess close to the ground-truth rather than penalizing a guess that deviates by a small number of pixels. During the training, network weight w and bias b are learned in predicted heatmaps $h_{i}\left(p;w,b\right)$.
(1)$${\hat{p}}_{i}=\arg\max_{p}h_{i}\left(p;w,b\right).$$

The output of a network will now be a continuous probability distribution on an input image plane, making it easier to see where the network’s guess is confident; in contrast, having a single position as an output does not show how the network is guessing.

Our goal in the first part of the network is to obtain the output feature maps that contain sufficient pixel-level details, high-resolution outputs that remain the same size as the input image (no resolution loss) and less extensive computation. A FCNs-based heatmap regression, followed by a kernel convolution, is used to meet our goal. To do so, we initially transform the facial landmarks’ ground-truth location $p_{i}^{gt}\left(x,y\right)$ of $i^{th}$ key point into target heatmap $h_{i}^{gt}\left(p\right)$ of $i^{th}$ key point (Figure 4a) via 2D Gaussian kernel (Equation (Equation 2)). Then, the target heatmap $h_{i}^{gt}\left(p\right)$ are fed into FC-DenseNets and finally convolved with a kernel convolution as illustrated in Figure 4b. In fully convolutional heatmap regression fashion, the task becomes one of predicting per-pixel likelihood of each key point’s heatmap from the image. It regresses the target heatmap of each landmark $h_{i}^{gt}\left(p\right)$ directly to obtain the response map M(p) stated in Equation (Equation 3), which has the same resolution as the input image.

We transform ground-truth location $p_{i}^{gt}\left(x,y\right)$ to target heatmap $h_{i}^{gt}\left(p\right)$ as
(2)$$h_{i}^{gt}\left(p\right)=\frac{1}{\sigma \sqrt{2\pi }}\exp\left(-\frac{∥p-p_{i}^{gt}∥}{2\sigma^{2}}\right),p\in \Omega ,$$
where σ is the standard deviation for the heatmaps used to control the response scope and Ω is the set of all pixel locations in image *I*.

We set the FC-DenseNet architecture to include 56 layers following Reference [19], which had FC-DenseNet56 with 4 layers per dense block and growth rate = 12. We adopted the smallest FC-DenseNet to reduce network computational complexity, as shown in Table 1, while still achieving notable outcomes compared with current popular architectures. We also applied fully convolutional ResNets with 50 layers (FC-ResNets50 [41]), available in the PyTorch framework [42] (Torchvision) and then compared the outcomes with fully convolutional DenseNets with 56 layers (FC-DenseNet56). As expected, FC-DenseNets56 outperformed FC-ResNets50 due to more depth and hence more parameters.

#### Kernel Convolution

The output of FC-DenseNets is in a channel-wise fashion that has the same resolution as the input image. After reaching the output resolution of the network, an implicit 45×45 pixel kernel convolution $K_{\sigma }$ is applied to produce a clear shape output of the feature maps. For computational efficiency, the kernel convolution $K_{\sigma }$ was generated by the Gaussian function in Equation (Equation 2). Here, the kernel convolution filter acts as a point-spread function to blur the input feature maps as shown in Figure 5. The kernel convolution filter $K_{\sigma }$ removes the detail and noise and provides gentler smoothing by preserving the edges of the feature maps. Without the kernel convolution, landmarks’ sub-pixel positions are neglected [43].

The kernel convolution filter convolves with the entire image using grouped convolution [44], which allows for more efficient learning and improved representation. In grouped convolutions, each input channel is convolved with its own filter. The final output of the network is a set of heatmaps that contain the probability of each key point’s presence at each pixel. With the convolved response maps $M\left(p\right)=\left[h_{i}^{gt}\left(p\right)|i=1\ldots N\right]$ and a kernel convolution filter $K_{\sigma }$, we can obtain the density heatmap $H^{0}$ as follows: (3)$$H^{0}=M\left(p\right)*K_{\sigma }$$

### 3.2. Dilated Skip Convolution Network for Shape Refinement

To enable networks to learn the spatial relationships between each key point and make better guesses, it must be able to view large portions of the input images. The portion of the input image viewed by the network is called the receptive field. Using the vanilla convolution filter [45] is a challenge when using a large receptive field: it is computationally expensive and can be easily overfitted due to the vast number of parameters. This problem is usually tackled by using pooling layers in conventional CNNs. Pooling layers choose one pixel from its field and discard other information, thereby reducing information and resolution of the input image. This degrades the performance of the network because some important information is lost when the resolution is decreased. Fortunately, dilated convolutions [37] solve this problem by using sparse kernels to alternate the pooling and convolutional layer, which dilates the kernels with zeros as a result of not only affecting the number of parameters but also increasing the size of the receptive field. In practice, kernels with different dilation factors are convoluted to the input and the outputs of those kernels are concatenated for subsequent layers [9]. Subsequent layers have no missing information from the input image and fewer parameters with different receptive fields. To apply this concept, References [18,46] introduced a stack of dilated convolutions in their network that can enlarge the receptive field exponentially while keeping the number of parameters low. Inspired by this design, we constructed a dilated skip convolution network that combined seven consecutive zero-padded dilated convolutions and skip-connections to overcome the issue of scale variations. In the network, our dilation factors ranged from *d* = 1 to *d* = 32 as stated in Table 2.

This module was carefully designed to increase the performance of our dense prediction architecture and ensure accurate spatial information by aggregating multi-scale contextual information. Our objective was to combine intermediate feature representations to learn global-context information and improve the final heatmap predictions. We exploited dilated convolutions to extract the global-context from input feature maps and then progressively updated the initial heatmap $\left(H^{0}\right)$. Due to the capacity to capture texture information at the pixel level, concatenating dilated convolutions of sub-layers together aids the network-extracting features from different scales concurrently. We also built extra skip-connections and embedded them in our dilated convolutions network to add global information from the entire image to common knowledge of the network from the previous feature map ($H_{\tau -1}\left[.\right]$). During the training, skip-connections concatenated output feature maps from previous and current layers together. Thus, our dilated skip convolution’s feature map $H^{DSC}\left[.\right]$, which has current feature map $H_{\tau }$, previous feature map $H_{\tau -1}\left[.\right]$, kernel filter k[.] and dilation factor *d*, is defined as:(4)$$H^{DSC}\left[x,y\right]=\sum_{i}\sum_{j}k\left[i,j\right]\cdot H_{\tau }\left[x-di,y-dj\right]+H_{\tau -1}\left[x,y\right].$$

Intuitively, Equation (Equation 4) shows that the model learns from each dilated convolution layer and the input initial heatmap, $H^{0}$, providing robustness against appearance changes. This is achieved through skip connections, which are extra connections between $H^{0}$ and dilated layers with different dilation factors, *d*. Consider the output feature map for the $n^{th}$ layer, $H^{n}$ and a non-linear transformation of the $n^{th}$ layer, $T_{n}\left(.\right)$. At each stage, the kernel, k[.], convolves with $H^{n}$ and then concatenates it with $H^{0}$. Thus, from Table 2, the network from the initial to the final output feature map for a DSC subnet with 7 dilation factors can be formulated as
(5)$$\left\{\begin{array}{c} H^{1}=T_{1}\left(H^{0}\right)\\ H^{2}=T_{2}\left(\left[H^{0},H^{1}\right]\right)\\ \vdots \\ H^{7}=T_{7}\left(\left[H^{0},\ldots ,H^{6}\right]\right), \end{array}\right.$$
where [$H^{0},H^{n}$] donates feature map concatenation.

Rather than having the dilated skip convolution network predicting the landmark locations from scratch in Equation (Equation 6), it is beneficial to refine the LAI subnet predictions. This was achieved by summing $H^{0}$ and $H^{DSC}$ to obtain the final feature map of the architecture,
(6)$$H^{f}=H^{0}+H^{DSC}.$$

To better understand how the heatmap is regressed in a real image, we transferred back $H^{0}$, $H^{DSC}$, $H^{f}$ to $S^{0}$, $S^{DSC}$, $S^{f}$. Thus, Equation (Equation 6) was replaced as follows: (7)$$S^{f}=S^{0}+S^{DSC}.$$

Figure 6 compares visualizations of landmark coordinates (green dots) in the real face image for both stages. Landmark coordinates from Figure 6a are improved in the second stage, for example the green dots with red circles in Figure 6b locate more correctly on the face contour and there is no missed landmark on the left eyebrow compared to Figure 6a.

Thus, dilated convolutions offer a method to increase global view exponentially on input image, hence the dilation factors should be set as exponential values following [35],
(8)$$d_{\left(i+1\right)}=2^{i},\;\mathrm{for}\;i=\left(0,1,2,\ldots ,n-2\right),$$
where $d_{\left(i+1\right)}$ is the dilation factor for the ${\left(i+1\right)}^{th}$ layer and *n* is the number of layers. In this case, the dilated convolution has 7 layers, hence optimal dilation factors $d_{\left(i+1\right)}\leq 32$, for i=(0,1,…,5). Table 2 shows dilation factors = 1, 1, 2, 4, 8, 16, 32, where the first two layers serve as conventional convolution layers.

Table 2 compares the proposed method’s using the mean error rate of the datasets, which should ideally be as small as possible. Thus, we need to find the optimal number of dilated layers most suitable for our entire network. Table 2 shows the optimal number of dilated layers = 7. Increasing the number of layers beyond that does not significantly improve the mean error rate, while introducing more parameters for the network and aggressively widening the receptive field via dilation factors would be detrimental to local features of small objects.

**Algorithm 1** Dilated skip convolution for facial landmark detection

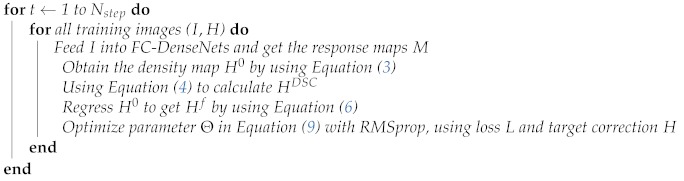



## 4. Experiments

### 4.1. Datasets and Data Augmentation

#### 4.1.1. Datasets

To evaluate the proposed algorithms, various datasets were created to investigate the robustness of the algorithms for imitating landmark detection in real-life situations. The datasets contained independent variations in pose, expression, illumination, background, occlusion and image quality. For instance, the 300W dataset [22] consisted of a wide range of head pose images and AFLW2000-3D [43] contained large-scale images in 3D. For training and validation, we used 300W-LP [23], a synthetically expanded version of 300W, as a basis to train our model. The model was fine-tuned with LFPW, HELEN and 300W datasets. To observe how the network was flexible with unseen datasets, we analyzed the AFLW2000-3D dataset without training it in advance, as presented in Table 3. In our evaluation experiments, we implemented our proposed algorithm (Algorithm 1) in “in-the-wild” datasets as follows:300W-LP [23]: 300W Large Pose (300W-LP) dataset consists of 61,225 images with 68 key points for each facial image in both 2D landmarks and the 2D projections of 3D landmarks. It is a synthetically-enlarged version of the 300W for obtaining face appearance in larger poses.LFPW [20]: The Labeled Face Parts in-the-Wild (LFPW) dataset has 1035 images divided into two parts: 811 images for training and 224 images for testing.HELEN [21]: HELEN consists of 2000 training and 330 test images with highly accurate, detailed and consistent annotations of the primary facial components. It uses annotated Flickr images.300W [22]: The 300 faces in-the-Wild (300W) dataset consists of 3148 images with 68 annotated points on each face for training sets collected from three wild datasets such as LFPW [20], AFW [47] and HELEN [21]. There are three subsets for testing: challenging, common and full set. For the challenging subset, we collected the images from iBUG [48] dataset which contains 135 images; for the common subset, we collected 554 images from the testing sets of HELEN and LFPW datasets; for the full set subset, we merged the challenging and common subsets (689 images).AFLW 2000-3D [23]: Annotated Facial Landmarks in the Wild with 2000 three-dimensional images (AFLW 2000-3D) is a 3D face dataset constructed with 2D landmarks from the first 2000 images with yaw angles between ±90${}^{{}^{\circ}}$ of AFLW [49] samples. It varies expression and illumination conditions. However, some annotations, especially larger poses or occluded faces, are not very accurate.

#### 4.1.2. Data Augmentation

For data augmentation (e.g., randomly flipping, resizing and cropping images, etc.), PyTorch framework [42] leaves the original input images untouched, returning only a changed copy at every batch generation.

To reduce overfitting in our model, we artificially expanded the amount of training data using random augmentation including cropping, rotation, flipping, color jittering, scale noise and random occlusion. We rotated the input image with a random angle of ±50${}^{{}^{\circ}}$ and scale noise from 0.8 to 1.2. We also scaled the longest side to 256 resulting in a 256×H or H×256 image, where H≤256.

### 4.2. Experimental Setting

#### 4.2.1. Implementation Detail

We implemented our model based on the open source PyTorch framework [42], which is a dynamic program that runs on a GPU. First, we cropped an input image to 256×256 resolution and generated an output set of response maps with the same resolution. Then, we transferred the image’s facial key points to heatmap key points using the 2D Gaussian kernel. In our method, the variance (sigma) of the 2D Gaussian kernel in the ideal response map was set to 0.25. For training, we optimized the network parameters by RMSprop [50] with a momentum of 0.9 and a weight decay of $10^{-4}$. We trained our model for 100 epochs with an initial learning rate of $10^{-4}$. We reduced it subsequently to $10^{-5}$ after 50 epochs and to $10^{-6}$ after another 80 epochs.

For loss function in our network, we chose the Euclidean distance loss function for our network,
(9)$$L\left(\Theta \right)=\frac{1}{N}\sum_{i=1}^{N}{∥Z\left(X_{i};\Theta \right)-Z_{i}^{gt}∥}_{2}^{2}$$
where *N* is the size of the training batch and $Z(X_{i},\Theta )$ is the output generated by the DSC network with parameters shown as Θ. $X_{i}$ represents the input images and $Z_{i}^{gt}$ is the ground-truth result of input image $X_{i}$.

During training, L(Θ) calculates the difference between the estimated and corresponding ground-truth feature map to update weight parameter Θ, to ultimately identify a set of parameters that make L(Θ) as small as possible.

#### 4.2.2. Evaluation

We evaluated for accuracy with three popular metrics: the normalized mean error (NME), the cumulative error distribution (CED) curve and the area under the curve (AUC). The NME was evaluated by measuring the distance between the detected landmark coordinates and the ground-truth facial landmark coordinates. It calculates the mean of the inter-pupil distance of multiple images which can be represented by
(10)$$\mathrm{NME}=\frac{1}{n}\sum_{i=1}^{n}\frac{∥xi-x_{i}^{gt}{∥}^{2}}{d},$$
where $x^{i}$ is the predicted coordinates and $x_{i}^{gt}$ is the ground-truth coordinates for $i^{th}$ image, *d* donates inter-ocular distance (Euclidean distance between two eye centres) and *n* is the total number of facial landmarks.

The CED is the cumulative distribution function of the normalized error which is larger than *l* and is reported as a failure. Thus, CED at the error is defined as
(11)$$\mathrm{CED}=\frac{N{}_{NME\leq l}}{n},$$
where $N_{\mathrm{NME}}$ is the number of images in which the error $NME_{i}$ is no higher than *l*.

AUC calculates the percentages of images that lie under certain thresholds. It is defined as:(12)$$AUC_{\alpha }=\int_{0}^{\alpha }f\left(e\right)de,$$
where *e* is the normalized error, f(e) is the CED function and α is the upper bound used to calculate the definite integration.

In this study, we present our evaluations using mean error rate and CED curves. We calculated additional statistics from the CED curves such as the AUC which is up to an error of 0.07. CED curves for our experiments on the 300W and AFLW2000-3D testing sets are illustrated in Figure 7. Furthermore, as clearly stated in the figure, the AUC of 300W dataset is 72.49% and 65.99% for AFLW200-3D dataset.

### 4.3. Comparison with State-of-the-Art Algorithms

#### 4.3.1. Comparison with LFPW Dataset

The goal of the LFPW dataset was to study the problem of unconstrained face conditions that were trained on 811 images and tested on 224 images. Images were collected from Google, Flickr and Yahoo using text queries.

Comparisons of different methods versus the proposed method are listed in Table 4. Our proposed method substantially reduced the mean error rate. The second-best mean error rate in the table is the CFSS [51] method, which has a mean error of 4.87%. Our method is considerably superior with an error rate of only 3.52%. Furthermore, compared to the SDM [5] method, which uses cascaded regressions and has an error rate of 5.67%, our method also prevails by 2.15%.

#### 4.3.2. Comparison with HELEN Dataset

Similar to the LFPW dataset, images were taken under unconstrained conditions with high resolutions and collected from Flickr using text queries. The dataset contained 2000 images for training and 330 images for testing.

Mean error comparisons of different methods on the HELEN dataset are presented in Table 5. Our method successfully achieved the lowest mean error percentage among all mentioned methods, with a mean error rate of 3.11% compared to the second-best, TCDCN [55], which achieved only a 4.60% error rate.

#### 4.3.3. Comparison with 300W Dataset

The 300W is an extremely challenging dataset that is widely used to compare the performance of different algorithms for facial landmark detection under the same evaluation protocol. Table 6 presents the comparison results of the mean error rate of the 300W dataset. Our method reduced the mean error rate by 3.60%, 8.69% and 3.90% for the common subset, challenging subset and full set subset. Moreover, our proposed method performed significantly better than the previous methods in full set subsets with an error reduction of 0.46% when compared to the second-best method, CPM [56]. Our method for common, challenging and full set subsets also demonstrated significant improvement compared to the current state-of-the-art method DeFA [57]. Its error rate was 5.37% for the common subset, 9.38% for the challenging subset and 6.10% for the full set subset, which are higher than in our proposed method. The example landmark detection results of our method are illustrated in Figure 8, which is a collection of example results from the common, challenging and full set subsets.

#### 4.3.4. Comparison of the AFLW2000-3D Dataset

The goal of the AFLW2000-3D dataset is to evaluate the algorithms on a large-pose dataset. In this dataset, we compared our proposed method with several state-of-the-art methods as presented in Table 7. The results show that our method had a mean error of 4.04%.

In comparison to 3DSTN [60], our method successfully reduced the mean error by 0.45% for the AFLW2000-3D dataset. The third best result in the dataset was DeFA [57], with an error rate of 4.50%. Our method has significantly and effectively improved errors in the dataset. The example landmark detection results of our method are illustrated in Figure 9.

## 5. Conclusions

In this paper, we presented a deep heatmap regression approach for facial landmark detection. We employed FC-DenseNets to extract dense feature maps along with an explicit kernel convolution for early-stage facial shape prediction. Starting with a suitable shape in the first stage, the detected shapes were refined to match the ground-truth shape during the last stage of the architecture. Our local appearance initialization subnet pursued a heatmap regression approach convolved with kernel convolution to serve as a local detector of facial landmarks in the first stage and the dilated skip convolution subnet was carefully designed to increase the performance of our dense prediction architecture and accurate spatial information by aggregating multi-scale contextual information for the sake of refining the local prediction of the first subnet. The proposed method achieved superior, or at least comparable, performance in comparison to state-of-the-art methods for challenging datasets, including LFPW, HELEN, 300W and ALFW2000-3D.

## Figures and Tables

**Figure 1 sensors-19-05350-f001:**
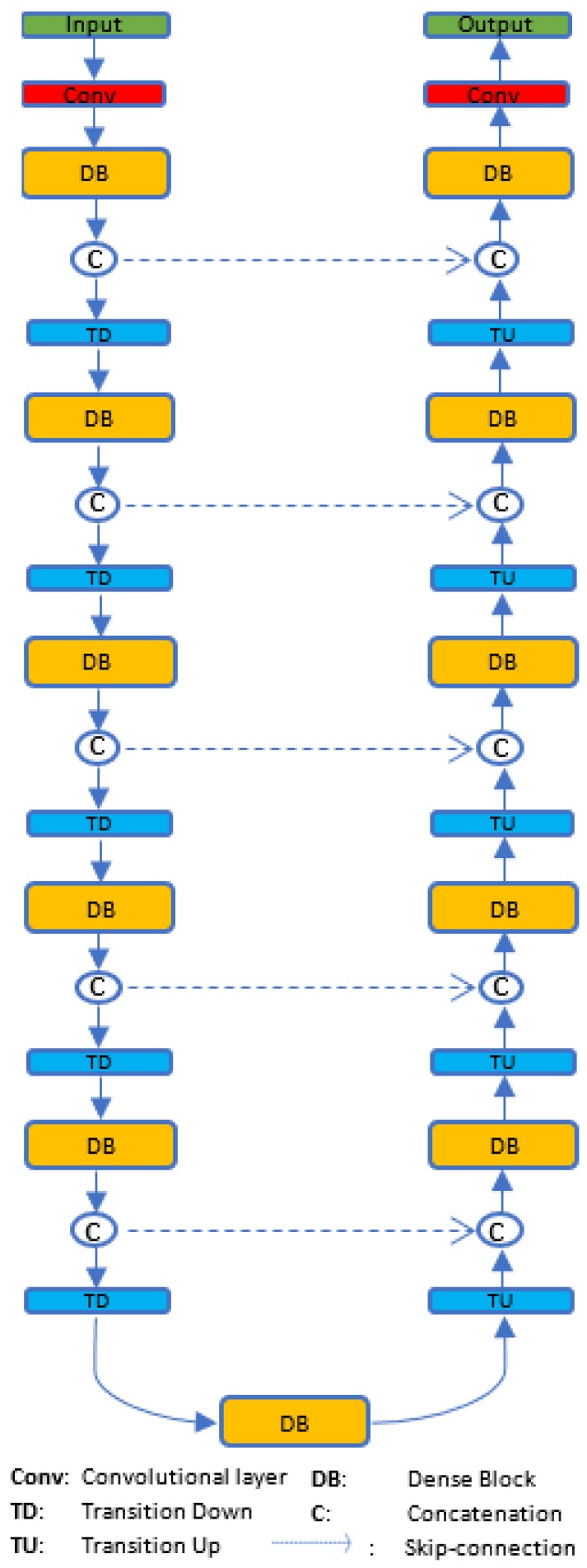
FC-DenseNet architecture.

**Figure 2 sensors-19-05350-f002:**
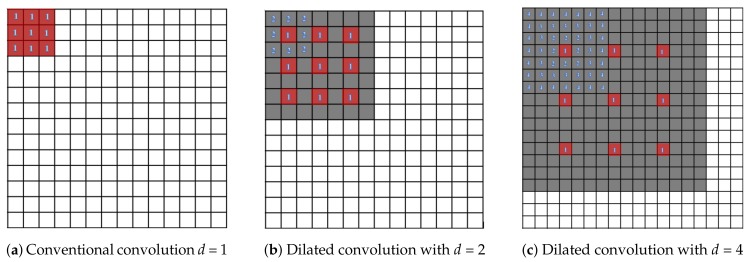
Conventional convolution and dilated convolution.

**Figure 3 sensors-19-05350-f003:**
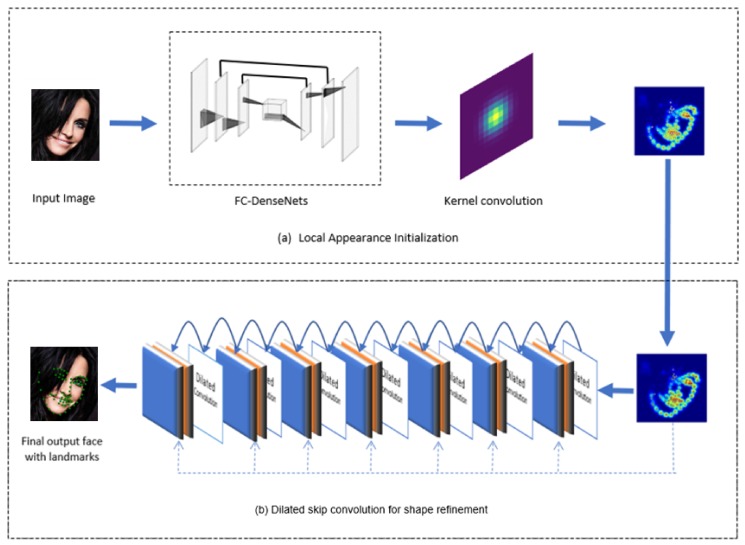
Overview of the proposed approach for facial landmark detection.

**Figure 4 sensors-19-05350-f004:**
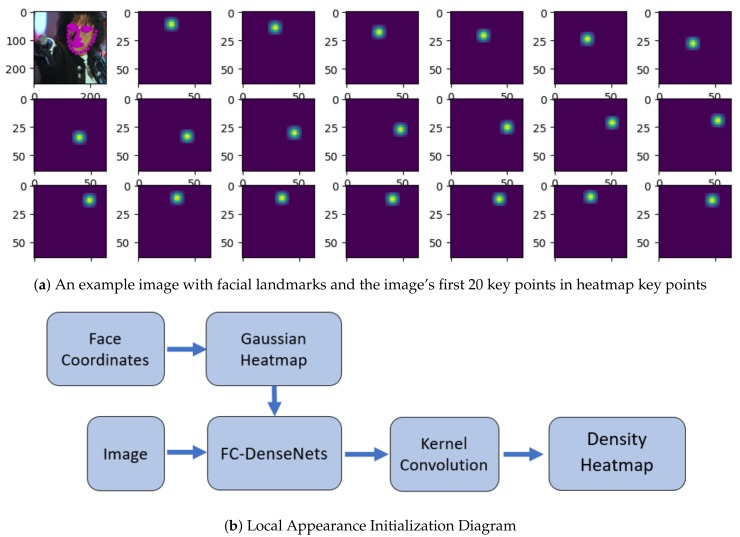
Local appearance initialization network.

**Figure 5 sensors-19-05350-f005:**

Best viewed in color. **Left**: Output of FC-DenseNets. **Middle**: Visualization of kernel convolution filter ($K_{\sigma }$). **Right**: Feature map after applying the filter ($K_{\sigma }$).

**Figure 6 sensors-19-05350-f006:**
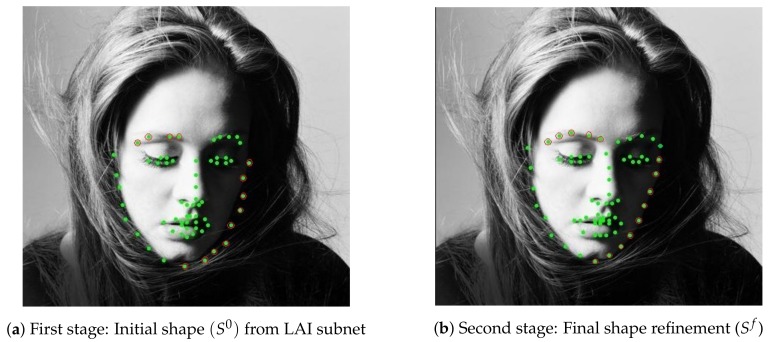
Dilated skip convolution network for shape refinement.

**Figure 7 sensors-19-05350-f007:**
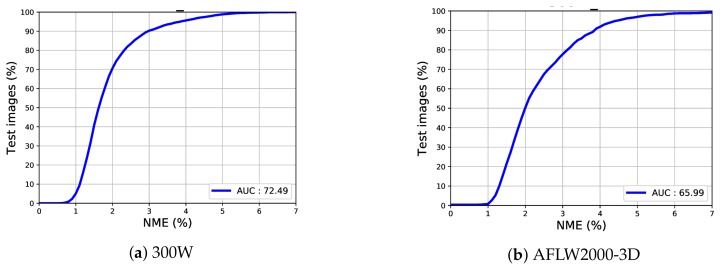
Cumulative error distribution (CED) curve and area under the curve (AUC).

**Figure 8 sensors-19-05350-f008:**
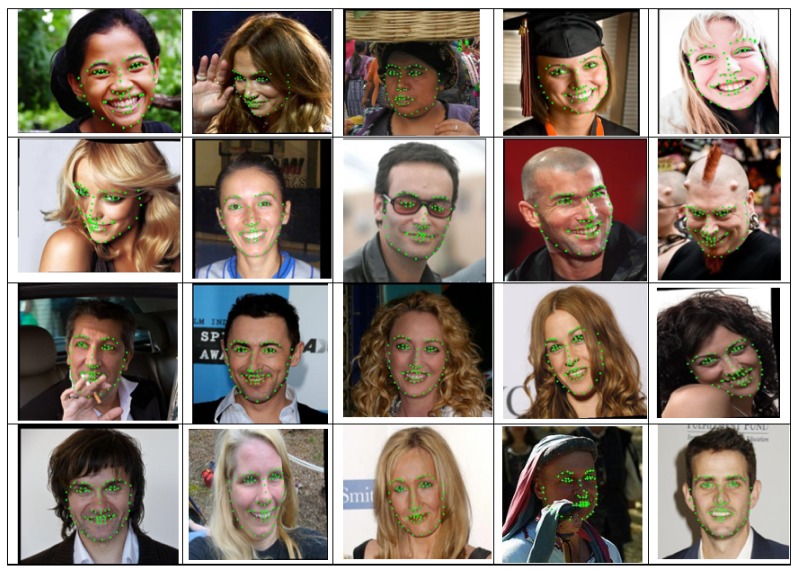
Landmark detection examples from the 300W dataset.

**Figure 9 sensors-19-05350-f009:**
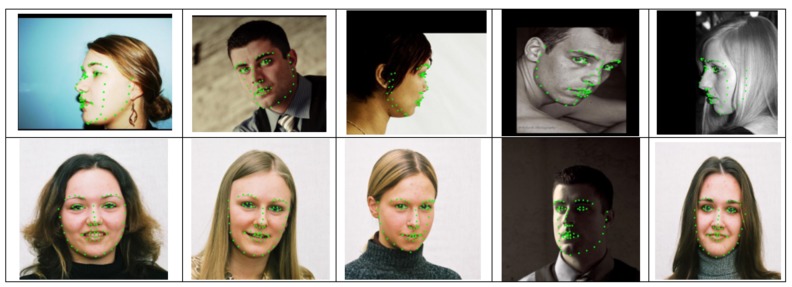
Landmark detection examples from AFLW2000-3D dataset.

**Table 1 sensors-19-05350-t001:** Architecture of FC-DenseNet56 used in the LAI network.

Layer	Number of Feature Maps
Input	3
3 × 3 convolution	36
DB (4 layers) + TD	84
DB (4 layers) + TD	144
DB (4 layers) + TD	228
DB (4 layers) + TD	348
DB (4 layers) + TD	492
DB (4 layers)	672
DB (4 layers) + TU	816
DB (4 layers) + TU	612
DB (4 layers) + TU	434
DB (4 layers) + TU	288
DB (4 layers) + TU	192
1 ×1	68 (keypoints)

**Table 2 sensors-19-05350-t002:** Structure of dilated convolutions.

Filter Size	Dilation Factor	Activation Function
3 × 3	*d* = 1	ReLU
3 × 3	*d* = 1	ReLU
3 × 3	*d* = 2	ReLU
3 × 3	*d* = 4	ReLU
3 × 3	*d* = 8	ReLU
3 × 3	*d* = 16	ReLU
3 × 3	*d* = 32	ReLU

**Table 3 sensors-19-05350-t003:** The list of face datasets used for training and testing.

Dataset	Landmark	Pose	Image
Training			
HELEN	68	±45${}^{{}^{\circ}}$	2000
LFPW	68	±45${}^{{}^{\circ}}$	811
300W	68	±45${}^{{}^{\circ}}$	3148
300W-LP	68	±90${}^{{}^{\circ}}$	61,225
Testing			
HELEN	68	±45${}^{{}^{\circ}}$	330
LFPW	68	±45${}^{{}^{\circ}}$	224
300W	68	±45${}^{{}^{\circ}}$	689
AFLW2000-3D	68	±90${}^{{}^{\circ}}$	2000

**Table 4 sensors-19-05350-t004:** Mean error in LFPW dataset.

Method	68 pts
Zhu et al. [52]	8.29
DRMF [53]	6.57
RCPR [43]	5.67
SDM [5]	5.67
GN-DPM [54]	5.92
CFAN [25]	5.44
CFSS [51]	4.87
CFSS Practical [51]	4.90
Ours	3.52

**Table 5 sensors-19-05350-t005:** Mean error on HELEN dataset.

Method	68 pts
Zhu et al. [52]	8.16
DRMF [53]	6.70
ESR [11]	5.70
RCPR [43]	5.93
SDM [5]	5.50
GN-DPM [54]	5.69
CFAN [25]	5.53
CFSS [51]	4.63
CFSS Practical [51]	4.72
TCDCN [55]	4.60
Ours	3.11

**Table 6 sensors-19-05350-t006:** Mean error on 300W dataset.

Method	Common	Challenging	Fullset
RCPR [43]	6.18	17.26	7.58
SDM [5]	5.57	15.40	7.50
LBF [58]	4.95	11.98	6.32
CFSS [51]	4.73	9.98	5.76
CFSS Practical [51]	4.79	10.92	5.99
RAR [59]	4.12	8.35	4.94
3DDFA [23]	6.15	10.59	7.01
DeFA [57]	5.37	9.38	6.10
CPM [56]	3.39	8.14	4.36
Ours	3.60	8.69	3.90

**Table 7 sensors-19-05350-t007:** Mean error on AFLW2000 dataset.

Method	68 pts
ESR [11]	7.99
RCPR [43]	7.80
MDM [61]	6.41
SDM [5]	6.12
3DDFA [23]	5.42
3DSTN [60]	4.49
DeFA [57]	4.50
Ours	4.04

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
