# Peer review of "Dilated Skip Convolution for Facial Landmark Detection"

_sensors, 2019, doi:10.3390/s19245350_

Round 1

Reviewer 1 Report

Many studies have been proposed and implemented to deal with the problems of localizing facial landmarks from given images. In this work, the authors consider facial information within both global and local contexts. They aim is to obtain local pixel-level accuracy for local-context information in the first stage and integrate this with knowledge of spatial relationships between each key point in a whole image for global-context information in the second stage. The proposed architecture consists of two main components, namely: a deep network for local-context subnet that generates detection heatmaps via fully convolutional Dense-Nets, and a dilated skip convolution subnet. The test had been carried out on some datasets (including LFPW, HELEN, 300W, and AFLW2000-3D), and the results are quite good.

General comments

 The novelty of the paper is quite limited.

The description given in the paper is good.

The results appear to be quite good.

Missing important references, related to Head Pose Estimation and related databases.

Improve SoA reference for face recognition (the one provided in [2] is updated).

e.g. substitute with https://ieeexplore.ieee.org/abstract/document/7374704 (PAMI 2016)

IEEE TRANSACTIONS ON PATTERN ANALYSIS AND MACHINE INTELLIGENCE, VOL. 38,

NO. 8, AUGUST 2016

Survey on RGB, 3D, Thermal, and Multimodal Approaches for Facial Expression Recognition:

History, Trends, and Affect-Related Applications

Ciprian Adrian Corneanu, Marc Oliu Simon, Jeffrey F. Cohn, and Sergio Escalera Guerrero

Improve SoA reference for face modelling (the one provided in [4] is updated). e.g. substitute with https://www.sciencedirect.com/science/article/pii/S0031320317303527 (PR2018)

Gaussian mixture 3D morphable face model

PaulKoppenaZhen-HuaFengabJosefKittlera

MuhammadAwaisaWilliamChristmasaXiao-JunWubHe-FengYinb

Pattern Recognition

Volume 74, February 2018, Pages 617-628

Include in SoA other papers on face pose and related DBs (beyond [19]), e.g. https://www.sciencedirect.com/science/article/pii/S0923596518302169  

and https://www.sciencedirect.com/science/article/pii/S235234091930232X?via%3Dihub 

(both from 2019)

FASSEG: A FAce semantic SEGmentation repository for face image analysis

S. Benini, K. Khan, R. Leonardi, M. Mauro, P. Migliorati

Data in brief.

Face analysis through semantic face segmentation

S. Benini, K. Khan, R. Leonardi, M. Mauro, P. Migliorati

Signal Processing: Image Communication.

Eq. 4,5,6,7 should be better explained.

Fig. 6 should be better explained and commented.

Specific comments

Enlarge Fig.1, Fig.3.

Check the consistency of Eq. 2.

Reviewer 2 Report

This paper considered facial information within both global and local contexts. The authors aim to obtain local pixel-level accuracy for local-context information in the first stage and integrate this with knowledge of spatial relationships between each key point in a whole image for global-context information in the second stage.

Please explain “Considerable recent researches have contributed to the development of models to deal with the location of key landmarks more accurately”. What is the advantage of deep learning compared to SIFT? In the related works, the author should expand the “regression-based methods”. Pls discuss related literatures, see “Intelligent facial emotion recognition based on stationary wavelet entropy and Jaya algorithm”, “Facial Emotion Recognition based on Biorthogonal Wavelet Entropy, Fuzzy Support Vector Machine, and Stratified Cross Validation”. Figure 1, what do the solid line and dashed line mean? What is the advantage of dilated convolution compared to ordinary convolution? How to set the optimal values of dilation? Table 1, how do you set the architecture of FC-DenseNet56? Did you test other pretrained models? Table 2, how do you assess your proposed method?

Round 2

Reviewer 1 Report

accept as it is.

Reviewer 2 Report

Accept